# β-Arrestin 2 as a Prognostic Indicator and Immunomodulatory Factor in Multiple Myeloma

**DOI:** 10.3390/cells14070496

**Published:** 2025-03-26

**Authors:** Parker Mathews, Xiaobei Wang, Jian Wu, Shaima Jabbar, Kimberly Burcher, Lindsay Rein, Yubin Kang

**Affiliations:** Division of Hematologic Malignancies and Cellular Therapy, Department of Medicine, School of Medicine, Duke University Medical Center, Durham, NC 27710, USA; parker.mathews@wustl.edu (P.M.); xiaobei@wustl.edu (X.W.); jian.wu@duke.edu (J.W.); shaimajabbar1@gmail.com (S.J.); kimberly.burcher@duke.edu (K.B.); lindsay.magura@duke.edu (L.R.)

**Keywords:** β-arrestin 2 (ARRB2), multiple myeloma, progression-free survival, overall survival, immune check points, G protein-coupled receptor (GPCR)

## Abstract

β-arrestin 2 (ARRB2) is involved in the desensitization and trafficking of G protein-coupled receptors (GPCRs) and plays a critical role in cell proliferation, apoptosis, chemotaxis, and immune response modulation. The role of ARRB2 in the pathogenesis of multiple myeloma (MM) has not been elucidated. This study addressed this question by evaluating the expression of ARRB2 in bone marrow (BM) samples from newly diagnosed MM patients and deriving correlations with key clinical outcomes. In light of recent trends towards the use of immune checkpoint inhibitors across malignancies, the effect of ARRB2 in the regulation of the PD-1/PD-L1 axis was also investigated. The expression of ARRB2 was significantly higher in MM patients resistant to proteosome inhibitor (bortezomib) treatment compared to those who responded. Higher ARRB2 expression in the BM of newly diagnosed MM patients was associated with inferior progression-free survival and overall survival. PD-1 expression was downregulated in CD3 T cells isolated from ARRB2 knockout (KO) mice. Furthermore, knockdown of ARRB2 with siRNA reduced PD-1 expression in murine CD3 T cells and PD-L1 expression in murine myeloid-derived suppressor cells. These findings suggest an important role of ARRB2 in MM pathogenesis, potentially mediated via modulation of immune checkpoints in the tumor microenvironment. Our study provides new evidence that ARRB2 may have non-canonical functions independent of GPCRs with relevance to the understanding of MM pathobiology as well as immunotherapy and checkpoint inhibitor escape/resistance more broadly.

## 1. Introduction

Multiple myeloma (MM) is a malignant plasma cell disorder with a high burden of morbidity and mortality and no current curative therapy. MM represents roughly 1% of all malignancies and 18% of hematologic malignancies in the United States [1,2] accounting for an estimated 35,730 new diagnoses and 12,590 deaths in 2023. Over the last several decades, the development and FDA approval of several active agents in multiple drug classes for the treatment for MM has led to a dramatic improvement in the depth and duration of clinical responses as well as the increased overall survival (OS) of patients with MM [3]. The median survival of patients with newly diagnosed multiple myeloma (NDMM) now reaches 7–9 years, and a proportion of myeloma patients survive longer than 10 years [4]. Despite these improvements, MM remains an incurable disease, and nearly all MM patients will eventually relapse and develop resistance to currently available agents, highlighting the importance of further understanding the molecules and signaling pathways driving the pathogenesis of MM in both the newly diagnosed and relapsed/refractory settings.

β-arrestin 2 (ARRB2) belongs to the arrestin family, a group of proteins key to the desensitization and trafficking of G protein-coupled receptors (GPCRs) [5,6]. GPCRs are the largest and most diverse group of membrane receptors in eukaryotes and are broadly important targets in modern pharmacology and drug discovery [7]. There are four members of the arrestin family, including the two “visual” arrestins (arrestin 1 and 4) and two non-visual forms, β-arrestin-1 (ARRB1) and ARRB2 [8,9]. The term “visual” is derived from the canonical role the visual arrestins play in the rhodopsin pathway for biological light perception. β-arrestins bind to a wide variety of signaling proteins and sequester key signaling pathway intermediates away from potential regulators [10]. β-arrestins regulate over 800 known GPCRs, providing a unique final common pathway with pharmacological relevance to the countless GPCR-directed drugs [11,12]. β-arrestins are central in many physiological and pathophysiological processes, including cell proliferation [13], non-proliferative cell growth [14], cell survival and apoptosis [15,16], cell migration and chemotaxis [17,18], and modulation of the immune response [19,20]. In part due to a mechanistic link to cell growth, survival, and migration, perturbation of β-arrestin has been shown to contribute to a more aggressive cancer phenotype [21,22,23]. Diverse roles for arrestins have been reported in critical oncogenic pathways including Scr/MAPK, Wnt, NF-kB, and PI3K/AKT [24]. ARRB2 constitutive knockout (KO) mice have been generated and are viable with a grossly normal phenotype; however, dual KO of ARRB2 and ARRB1 is embryonically lethal, suggesting a compensatory and overlapping role in GPCR signaling [25,26]. ARRB2 has been implicated in the pathogenesis of chronic myeloid leukemia and primary myelofibrosis [27]. In myelofibrosis, mice who received transplanted hematopoietic stem cells (HSCs) from β-arrestin 2-KO donors did not develop myelofibrosis after exposure to MPLW515L-mutant retroviruses, while controls were uniformly affected [27]. The role of ARRB2 in MM pathogenesis has not been previously reported in the literature.

The programmed cell death protein 1 (PD-1)/programmed cell death-ligand 1 (PD-L1) pathway is a major negative regulator of the adaptive immune system and is leveraged to remarkable clinical benefit with immune checkpoint inhibitor therapy [28]. PD-1 and PD-L1 are members of type I transmembrane glycoproteins that consists of the immunoglobulin-like extracellular domain, transmembrane domain, and cytoplasmic tails and are essential for suppressing T cell activation and cytokine production [29]. PD-1 is expressed on the surface of antigen-activated T and B cells [30]. PD-L1 is expressed on several cell types, including resting T and B cells, dendritic cells, macrophages, vascular endothelial cells, and pancreatic islet cells [31]. PD-L1 expression is higher in the plasma cells from MM patients compared with cells from MGUS patients and healthy volunteers, and its expression is upregulated at relapse or in the treatment refractory phase [32,33]. Immunotherapy targeting PD-1/PD-L1 immune checkpoints has resulted in unprecedented tumor responses and long-term survival benefits in patients with advanced solid malignancies and Hodgkin lymphoma [34,35,36,37]. However, checkpoint inhibitors have not demonstrated significant anti-myeloma activities as monotherapy in MM [38]. In fact, the combination of checkpoint inhibitors with immunomodulatory agents (IMiDs, lenalidomide and pomalidomide) worsened overall survival in MM patients and was felt to be overly toxic with a high burden of significant adverse events [39]. Understanding how PD-1/PD-L1 expression is regulated has important ramifications in developing more effective immunotherapies targeting this pathway for both solid tumors and hematologic malignancies.

Given the shared BM microenvironment crucial to the pathobiology of myeloid neoplasms, myeloproliferative disorders, and plasma cell disorders, we hypothesized that ARRB2 is similarly important in the development of multiple myeloma. In the current study, we examined the correlation between ARRB2 expression levels in a clinical cohort utilizing newly diagnosed MM bone marrow biopsy samples and matched clinical outcomes. In search of a possible mechanism for the correlations described herein, we determined the effect of ARRB2 in the regulation of the PD-1/PD-L1 checkpoints in key murine immune cells.

## 2. Materials and Methods

### 2.1. Animal and Clinical Study Approval

The studies involving human patient samples were performed in accordance with the ethical standards of the Duke University Institutional Review Board Committees (IRB) (IRB protocol number: Pro00088486. Approval date: 11 June 2017). All studies involving vertebrate animal models were conducted in accordance with Duke University Institutional Animal Care and Use Committee (IACUC)-approved procedures under protocol number: A074-23-03 (approval date: 4 July 2023).

### 2.2. Clinical Data Collection and Analysis

A single center, retrospective cohort study was performed. Forty-seven patients with NDMM who were treated at our institution between 2005 and 2015 and had initial diagnosis bone marrow biopsy samples available were included in the study. Patient data were collected by review of the patients’ electronic medical records and stored in a secure RedCap database (https://projectredcap.org/, access time 1 June 2024) (Vanderbilt University, Nashville, TN, USA). The treatment response was characterized using the International Myeloma Working Group (IMWG) treatment response criteria and classified as complete remission (CR), very good partial response (VGPR), partial response (PR), stable disease (SD), or progressive disease (PD) [40,41]. The International Staging System (ISS) stage and the cytogenetic risk were defined using IMWG criteria [40,41,42,43]. Progression-free survival (PFS) was defined as the duration from the initiation of treatment to first progression or death, whichever was earlier. OS was defined as the duration from the date of diagnosis of MM to the date of death or date of last follow-up at which the patient was known to be alive, with those alive censored at the date of last contact.

### 2.3. Gene Expression Profile and Clinical Data

The GSE9782 gene expression dataset was downloaded from the Gene Expression Omnibus (GEO) database and used for the analysis of ARRB2 expression in patients responsive to bortezomib treatment vs. patients resistant to bortezomib. Additional survival data were accessed based on the Genomic Scape database (http://www.genomicscape.com/, accessed on 1 June 2024).

### 2.4. Reagents

Recombinant mouse IL-2 was purchased from R&D systems, Minneapolis, MN, USA (Cat. #402-ML). Recombinant GM-CSF was purchased from BioLegend, San Diego, CA, USA (Cat. #713704). Recombinant IL-6 was purchased from Millipore Sigma, Burlington, MA, USA (Cat. #I9646). Immunofluorescent-labeled ARRB2 antibody (A2CT) was produced in house at Dr. Robert Lefkowitz’s lab (Duke University). The following antibodies were purchased from commercial sources as indicated: anti-mouse CD3 (Cat. #16-0032-82) and anti-mouse CD28 (Cat. #16-0281-82) from Thermo Fisher, Waltham, MA, USA; anti-mouse CD3-PE (Cat. #100308), anti-mouse Ly6G/Ly6C (Gr-1)—APC-Cy7 (Cat. #108424), anti-mouse CD11b-PE (Cat. #101208), anti-mouse Ly-6C-PE-Cy7 (Cat. #128017), anti-mouse Ly-6G-FITC (Cat. #127606), anti-mouse PD-1-FITC (Cat. #135214), and anti-mouse PD-L1-PerCP-Cy5.5 (Cat. #124333) from BioLegend; anti-mouse CD4-APC-Cy7 (Cat. #552051) and anti-mouse CD8-PE-Cy7 (Cat. #552877) from BD Biosciences (San Jose, CA, USA); and anti-mouse S1PR1-APC (Cat. #FAB7089A) from R&D Systems (Minneapolis, MN, USA).

### 2.5. Immunofluorescent Staining of Bone Marrow ARRB2 and Quantification

Archived paraffin-embedded BM biopsy samples from patients (*n* = 47) with NDMM were sectioned at 5 μm, and 2 consecutive sections per patient were used. One section was stained with immunofluorescent-labeled ARRB2 antibody (A2CT, 1:500) in addition to Hoechst (H3570, 1:1000, Invitrogen, Carlsbad, CA, USA). The second section was stained with nonimmune mouse IgG and rabbit IgG. Immunofluorescence was measured using a Zeiss laser-scanning microscope. ARRB2 expression was quantified using Imaris 8.1 software and normalized to IgG isotype control and Hoechst expression. ARRB2 expression index number is calculated using the following: (A2CT signal from sample − A2CT signal from IgG control)/DAPI as described [44]. ARRB2 expression is categorized as high expression if the index number is ≥1 or low expression if the index number is <1.

### 2.6. ARRB2 Knockout Mice

Constitutive ARRB2 KO mice were generated on a C57Bl/6J (CD45.2) background by Dr. Robert Lefkowitz (Duke University) [45]. Wild-type littermates and the ARRB2 KO mice were kindly provided by Dr. Robert Lefkowitz.

### 2.7. Isolation, Activation, and In Vitro Culture of Murine T Cells and MDSC Cells

Total CD3+ T cells were harvested from spleens of WT C57/BL6 mice (Jackson Laboratory, Bar Harbor, ME or bred in house) or ARRB2 KO mice. Briefly, animals were euthanized, and spleens were dissected. Total CD^3+^ T cells were isolated with a murine CD3+ T-cell isolation column (R&D Biosystems, Minneapolis, MN, USA) according to the manufacturer instructions. For T-cell activation, T cells were plated on six-well plates (Genessee Scientific, Morrisville, NC, USA) pre-treated with anti-mouse CD3 1:1000 *v*/*v* and anti-mouse CD28 1:200 *v*/*v* for 24 h. For T-cell culture, T cells were plated in modified RPMI (Corning, Glendale, AZ, USA) containing 10% *v*/*v* FBS (Gemini Bio, West Sacramento, CA, USA), 1% *v*/*v* penicillin/streptomycin (Thermo Fisher), 1 mM sodium pyruvate (Thermo Fisher), 50 nM beta-mercaptoethanol (Thermo Fisher), and recombinant mouse IL-2 (50 IU/mL).

To generate MDSCs, BM cells were harvested from C57Bl/6 mice by flushing femurs and tibias, and red blood cells were lysed using hypotonic ACK buffer. BM single cell suspension was cultured in vitro with GM-CSF (20 ng/mL) and IL6 (20 ng/mL) for 4 days [46]. Cells were then harvested and characterized using CD11b, Gr-1, Ly6C, and Ly6G antibody.

### 2.8. Flow Cytometry Analysis

For immune cell subset profiling, single cell suspensions were pre-incubated with Mouse Fc BlockTM (clone 2.4G2, BD Bioscience), then stained with fluorescently labeled antibody cocktail for 30 min at 4 °C. Flow cytometry was performed on a 2 laser (488 nm and 633 nm) BD FACSCanto II flow cytometer (BD Bioscience) and the data were analyzed using FlowJo 10.8 software (Treestar, Ashland, OR, USA). Data were analyzed with FlowJo (BD Biosciences). The percentage and mean fluorescence intensity for PD-1 and S1PR1 were determined in spleen cells of ARRB2-KO mice. Murine cell populations were identified as follows: T cell: CD3^+^; CD8^+^ T cell: CD3^+^CD8^+^; CD4^+^ T cell: CD3^+^CD4^+^; total myeloid-derived suppressor cells (MDSCs): CD11b^+^ Gr-1^+^ cells; granulocytic (G)-MDSCs: CD11b^+^Ly6G^+^Ly6C^lo^ cells; and monocytic (M)-MDSCs: CD11b^+^Ly6G^-^Ly6C^hi^ cells.

### 2.9. Immunoblotting

Total protein was extracted and quantified using the BCA assay (Thermo Scientific, Waltham, MA, USA). Thirty µg proteins of each sample were separated by 10% SDS polyacrylamide gels and transferred to the PVDF membrane. The membrane was blocked with 5% BSA buffer for one hour at RT and incubated with primary antibody overnight at 4 °C. The membrane was then probed with a horseradish peroxidase (HRP)-conjugated secondary antibody and developed using a Pierce ECL substrate (Pierce, Rockford, IL, USA).

### 2.10. siRNA Knockdown

ARRB2-specific siRNA oligonucleotides were synthesized and purchased from Dharmacon (Lafayette, CO, USA). Two different siRNA sequences targeting human (rat) β-arrestin-2 were as follows: 5′-GGACCGC(G)AAAGUGUUUGUG-3′ and 5′-CCAACCUCAUUGAAUUU(C)GA-3′. Activated CD3 T cells or MDSCs were grown to 70–90% confluency. The cells were then washed with PBS twice and incubated with ARRB2-specific siRNA suspended in lipofectamine for 24 h. The cells were then washed and cultured for an additional 48 h. PD-1 and PD-L1 expressions were measured by flow cytometry. S1PR1, a GPCR, was used as a positive control.

### 2.11. Statistical Analysis

An overview of patient demographics and treatment has been tabulated (Table 1). Kaplan–Meier analysis was used to determine the median OS and PFS. The chi-square test and logistic regression analysis were used to study variables deemed to be possibly predictive of response. For in vitro studies, values reported and shown in graphical displays represent the mean standard error of the mean, unless stated otherwise. Comparisons of the mean expressions across groups were performed using two sample *t*-tests. Based on the distribution within groups, we used a *t*-test with equal or unequal variances. For all comparisons, *p* < 0.05 was used to denote significance. Statistical analyses used GraphPad Prism v9 (GraphPad Software).

## 3. Results

### 3.1. ARRB2 Expression Is Upregulated in Patients with MM, and ARRB2 Overexpression Is Associated with Poor Survival: An In-Silico Analysis

Given the accumulating evidence relating ARRB2 to multiple aspects of tumor development, we first wanted to investigate the expression level of ARRB2 in patients with MM and the correlation between ARRB2 expression and clinical outcomes. We performed an in-silico analysis of the APEX trial Gene Expression Omnibus (GEO) microarray database (GSE9782). This microarray dataset used purified myeloma samples from relapsed patients enrolled in phase 2 and phase 3 clinical trials of bortezomib [47]. Compared to bortezomib-responsive MM patients, patients who did not respond to bortezomib treatment had a significantly higher level of ARRB2 expression (NR vs. R, *p*-value = 0.0183) (Figure 1A). Additionally, a higher level of ARRB2 expression correlated with a significantly shorter OS (458 days vs. 647 days, *p* = 0.019). These data demonstrated an important role of ARRB2 in MM pathogenesis with prognostic relevance to a current standard of care in MM therapy (Figure 1B).

### 3.2. Higher ARRB2 Expression Correlates with Disease Progression and Poor Survival in Newly Diagnosed Myeloma (NDMM) Patients

We performed a retrospective cohort study to investigate the impact of ARRB2 on the clinical outcomes of patients with NDMM. We measured the expression level of ARRB2 using immunofluorescent staining in the archived BM samples of NDMM patients. A total of 47 patients with NDMM were included in the study. The quantitative level of ARRB2 in each bone marrow biopsy sample was assessed using the ARRB2 expression index number: (A2CT signal from sample − A2CT signal from IgG control)/DAPI. ARRB2 expression is categorized as high if the index number is ≥1 or low if the index number is <1. Patient characteristics are summarized in Table 1. Consistent with our in-silico analysis, higher ARRB2 expression in BM was associated with a lower overall response rate (to front-line IMID and PI, generally) and inferior PFS and OS. The overall treatment response rate in NDMM patients who had high ARRB2 expression was 56%. In contrast, the overall response rate was 82% in NDMM patients who had low ARRB2 expression (*p* = 0.0293). PFS was 22 months in NDMM patients who had high ARRB2 expression compared to 30.13 months in NDMM patients who had low ARRB2 expression (*p* = 0.0385). Similarly, OS was 86.4 months in high ARRB2 NDMM patients compared to 107.3 months in low ARRB2 NDMM patients (*p* = 0.0301) (Figure 2). These data further demonstrate that ARRB2 expression is associated with PFS and OS in patients with NDMM, with potential prognostic implications.

### 3.3. ARRB2 Is Involved in the Regulation of PD-1 Expression in T Cells

The tumor microenvironment (TME) plays a critical role in the pathogenesis of MM. The PD-1/PD-L1 signaling pathway is central to tumor intrinsic and extrinsic immunosuppression and immune evasion. PD-1/PD-L1 can inhibit the activation of T lymphocytes and enhance the immune tolerance of tumor cells, thereby promoting tumor immune escape [48,49]. To assess the role of ARRB2 in the regulation of PD-1 expression, we isolated CD3 T cells from ARRB2 KO mice and the WT littermates. T cells were stimulated with CD3/CD28 antibodies. PD-1 and S1PR1 expression was measured by flow cytometry over time. CD3 T cells isolated from ARRB2 KO mice had significantly reduced PD-1 expression (Figure 3A). As a positive control, the expression of S1PR, a GPCR which would be canonically related to beta-arrestin, was significantly reduced in ARRB2 KO mice (Figure 3B).

To further elucidate the role of ARRB2 in the regulation of PD-1 expression, siRNA-mediated knockdown of ARRB2 was performed in murine T cells, with successful silencing confirmed by Western blot analysis (Figure 4A). Treatment with ARRB2-specific siRNA significantly reduced PD-1 expression in murine CD3^+^ T cells, CD4^+^ T cells, and CD8^+^ T cells. As anticipated, ARRB2 knockdown in CD3^+^ T cells also led to a reduction in surface S1PR1 expression (Figure 4B,C).

### 3.4. ARRB2 Regulates PD-L1 Expression in Myeloid-Derived Suppressor Cells (MDSCs)

PD-L1 on MDSCs is implicated in tumorigenesis and contributes to MDSCs-mediated T-cell suppression [50,51]. Tumor-infiltrating MDSCs express a higher level of PD-L1 than peripheral MDSCs in circulation or in the bone marrow [52]. Immune checkpoint inhibitor therapy reduces the suppressive capacity of MDSCs on T cells and has resulted in therapeutic control of the tumor in numerous disease states [53,54]. To determine the role of ARRB2 in the regulation of surface PD-L1 expression in MDSCs, we isolated bone marrow cells from mice and cultured with GM-CSF and IL6 to induce the differentiation of BM cells to MDSCs. MDSCs were characterized with CD11b, Ly6C, and Ly6G as described [46] and treated with ARRB2-specific siRNA. Silencing of ARRB2 in MDSCs led to a reduction in PD-L1 expression in murine MDSCs, granulocytic (G-) MDSCs, and monocytic (M-) MDSCs (Figure 5A), consistent with the decreased expression of S1PR1, a G protein-coupled receptor (GPCR) serving as a positive control (Figure 5B). Our data suggest that ARRB2 may contribute to the MDSC-mediated T cell regulation, consistent with the clinical outcomes data discussed above.

## 4. Discussion

In the current study, we performed in-silico analysis and immunofluorescent staining of bone marrow biopsy samples of NDMM patients and demonstrated that higher expression of ARRB2 correlated with bortezomib resistance and inferior PFS and OS. ARRB2 is a multifunctional intracellular protein that regulates the activity of many cellular signaling pathways and physiologic functions [55]. β-arrestins were discovered for their ability to disrupt signaling by binding to many activated GPCRs. Previous studies have shown that GPCR signaling can modulate progression, metastasis, treatment resistance, and other key oncogenic processes across many primary malignancies. [56,57]. A recent study demonstrated three major cellular mechanisms through which β-arrestins can promote cell cycle progression: Ras-mediated activation of mitogenic ERK1/2 signaling, transactivation of EGF receptors, and cytoskeletal [58]. In addition, many studies have shown that β-arrestins play a role in other facets of cancer initiation and progression [59,60,61]. Our study provides the first evidence linking β-arrestins to the pathogenesis of MM. Furthermore, our findings have important clinical relevance: the measurement of ARRB2 in bone marrow biopsy samples could help risk stratification of NDMM patients, and for patients with high ARRB2 expression, more intensive chemotherapy will be needed.

While it becomes clear that alteration of ARRB2 expression plays an important role in tumorigenesis, the effects of ARRB2 level in disease progression may vary depending on the tumor type. For instance, Sun et al. reported that in hepatocellular carcinoma, downregulation of ARRB2 promotes tumor invasion and correlates with poor disease outcome [23]. In contrast, in renal cell carcinoma and in ovarian cancer, it was found that overexpression of ARRB2 induces tumor growth and metastasis [21,22]. In hematologic malignancies, knockdown of ARRB2 was found to abrogate the development of primary myelofibrosis in a mouse model [27]. The discrepancy in the effects of ARRB2 level between hepatocellular carcinoma and renal cell cancer or ovarian cancer is unclear. This could be due to the differences between involved GPCRs, the subcellular location and function of ARRB2 [62,63,64,65,66], and the metabolic phenotypes of different cancers [65].

Our immunofluorescent staining in the BM biopsy samples of NDMM patients examined the overall expression level of ARRB2—we were unable to directly assess the cellular localization with greater granularity (for example, CD138+ myeloma cells versus BM microenvironment cells) due to constraints related to the archived bone marrow biopsy specimens. Additional evaluation of ARRB2 expression during treatment course and in relation to disease relapse will also be important to further define the role of ARRB2 in MM disease progression and treatment responses. We noted that fewer patients in the high ARRB2 expression group received autologous hematopoietic stem cell transplant (HSCT). However, we do not feel that this affects the prognostic significance of ARRB2 expression in NDMM patients. The MM IFM 2009 study [67] and the DETERMINATION Trial [68] demonstrated that although autologous HSCT prolongs PFS, it does not impact the OS. We indeed found that ARRB2 expression correlated with OS. Furthermore, the treatment response in the study was determined after the completion of induction therapy but prior to auto HSCT, and thus should not be affected by HSCT. Therefore, the correlation between ARRB2 expression and OS or treatment response following induction therapy was not due to the differences in the rate of auto HSCT. Further study with a larger patient cohort will be warranted to further validate our results.

To further dissect the roles of ARRB2 in the tumor microenvironment, we isolated CD3 T cells from ARRB2 mice and WT littermate mice and measured surface PD-1 expression by flow cytometry. We found that ARRB2 KO downregulated PD-1 expression in CD3 T cells. We performed confirmatory experiments using ARRB2-specific siRNA to validate the effects of ARRB2 in the regulation of PD-1 in T cells and PD-L1 in MDSCs. Our data clearly demonstrated that ARRB2 plays an important role in the regulation of the surface expression of PD-1/PD-L1 immune checkpoints.

PD-1/PD-L1 is the best characterized immune checkpoint. PD-1 on T cells binds to PD-L1, present on nonhematopoietic cells and cancer cells. This PD-1/PD-L1 interaction serves to dampen T-cell activity, which prevents autoimmune reactions in noncancerous conditions; however, it pathologically allows cancer cells to evade the immune system. The regulation of PD-1/PD-L1 expression is complex, involving regulation at the level of gene transcription, epigenetics, post-transcriptional and post translational modification, exosomal transport, and protein degradation [69,70]. PD-1/PD-L1 degradation is primarily mediated by the ubiquitin-mediated proteasome pathway. Our study provides the first evidence that ARRB2 is involved in the regulation of the surface expression of PD-1/PD-L1. Although β-arrestins are typically associated with GPCRs, recent studies have found that β-arrestins can also regulate the function of other transmembrane proteins beyond GPCRs including some type I transmembrane receptors and ion channels [71,72]. β-arrestins affect the expression and function of type I transmembrane receptors by mediating their internalization, scaffolding, and/or ubiquitination. Additional studies are needed to further clarify how ARRB2 is involved in these processes of PD-1/PD-L1. We postulated that ARRB2 regulates the recycling and/or the degradation pathways of PD-1 and PD-L1, thereby controlling their surface expression level. It is also expected that in addition to PD-1 and PD-L1, ARRB2 plays important roles in the regulation of many other molecules/receptors that are critical to MM pathogenesis.

Targeting β-arrestins for therapeutic purposes has been challenging due to complex signaling networks and numerous off-target effects. Additionally, β-arrestin expression varies across different tissues, raising the concern for a challenging side-effect profile. Experimental approaches have been developed to inhibit arrestin function by cell-specific genetic ablation or downregulation by siRNA, including aptamers (oligonucleotides with structures that allow selective binding to the surface of pathological target proteins and inhibit protein–protein interaction) [73]. Administration of an ARRB2-specific aptamer to leukemia cells has been shown to impair β-arrestin-dependent signaling and inhibit tumor progression in chronic myeloid leukemia models and primary human samples [74,75]. Inhibiting arrestin function would lead to two anticipated consequences: G protein-dependent signaling would be amplified due to the disruption of homologous desensitization, while signals generated by arrestin scaffolds would be diminished. Depending on the specific context, both effects can be therapeutically beneficial [66,76]. Biased agonists, small molecules that selectively activate beta-arrestin signaling pathways while minimizing G protein activation, have been proposed.

## 5. Conclusions

This study delves into the association between ARRB2 expression and clinical outcomes in patients with multiple myeloma (MM). It also explores the role of ARRB2 in regulating PD-1/PD-L1, an unexpected finding considering that PD-1 and PD-L1 are not GPCRs and would not be expected to be canonically regulated in this manner. Our study presents further evidence suggesting that β-arrestins may hold prognostic significance in newly diagnosed MM. This finding has significant implications for our understanding of MM pathogenesis and therapeutic responses, as well as the potential of immunotherapy and resistance to PD-1/PD-L1 blockade in a broader context.

## Figures and Tables

**Figure 1 cells-14-00496-f001:**
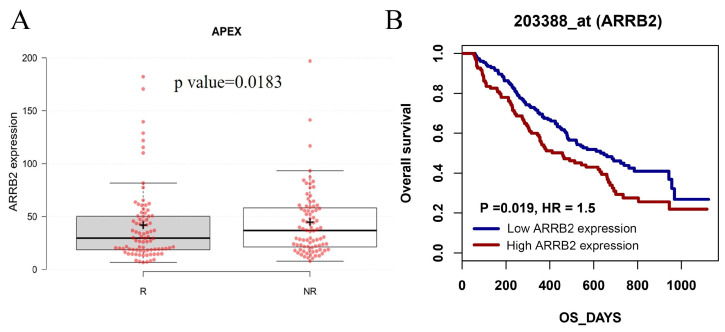
Association of ARRB2 expression with clinical outcomes in multiple myeloma patients. (**A**) ARRB2 expression levels in the bortezomib-responsive (R) and non-response (NR) groups were analyzed using microarray data from the APEX trial (GEO dataset GSE9782). Each red dot represents individual patient data. (**B**) Kaplan–Meier survival analysis illustrating overall survival (OS) of 528 multiple myeloma (MM) patients from the APEX clinical trial (GSE9782), stratified by βarr2 (ARRB2) expression levels. Low and high expression groups were defined as below and above the median AARB2 gene expression, respectively.

**Figure 2 cells-14-00496-f002:**
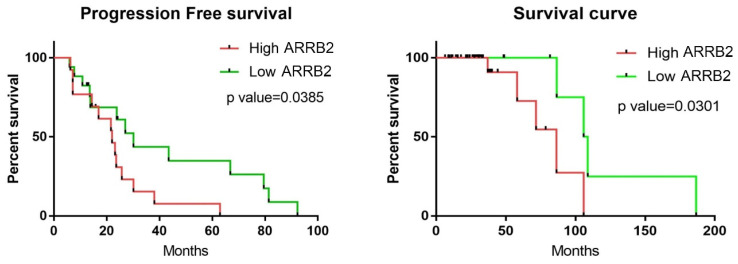
High βarr2 expression correlates with poor PFS and OS in multiple myeloma patients: retrospective cohort study. Kaplan–Meier survival curves showing progression-free survival (PFS) and overall survival (OS) for 47 newly diagnosed multiple myeloma (NDMM) patients in a retrospective cohort, stratified by ARRB2 expression levels. Low and high expression groups were defined by ARRB2 expression index number (low expression: index number < 1; high expression: index number ≥ 1).

**Figure 3 cells-14-00496-f003:**
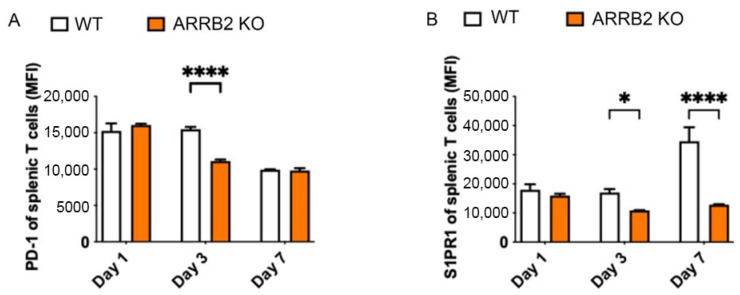
T cells from ARRB2 KO mice show downregulation of PD-1 and SIPR1 expression after ex vivo activation. T cells were harvested from ARR 2 KO and WT mice and activated in vitro with CD3 and CD28 antibodies. PD1 (**A**) and S1PR1 (**B**) expression levels were measured at selected time points by flow cytometry. * *p* < 0.05, **** *p* < 0.0001.

**Figure 4 cells-14-00496-f004:**
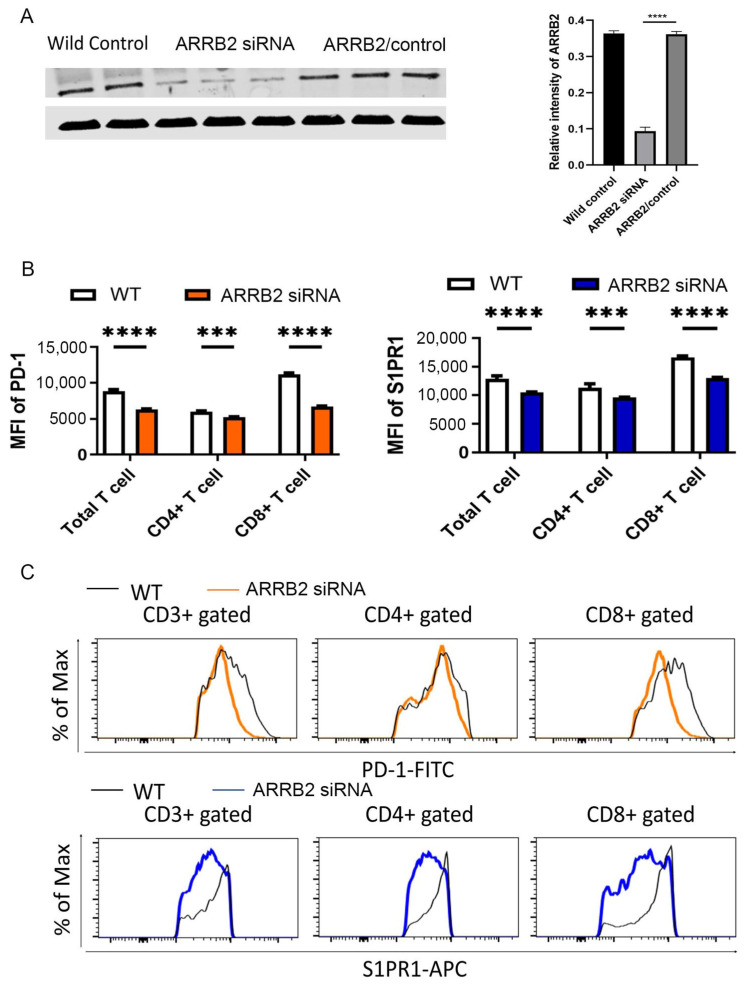
ARRB2 Knockdown reduces PD-1 expression in CD3, CD4, and CD8 T cells. (**A**, **left**) ARRB2 expression was silenced in murine splenic T cells using ARRB2-specific siRNA, and the knockdown efficiency was confirmed via Western blot analysis. (**A**, **right**) Relative intensity of ARRB2 expression was quantified using Image J software (version 1.41). (**B**) Splenic T cells and their subtypes subjected to ARRB2 siRNA knockdown displayed significantly reduced expression of PD-1 and S1PR1, as measured by flow cytometry assessing mean fluorescence intensity (MFI). (**C**) Representative flow cytometry histograms demonstrated reduced levels of PD-1 (orange) and S1PR1 (blue) in murine splenic T cells and their subtypes following ARRB2 siRNA-mediated knockdown. *** *p* < 0.001, **** *p* < 0.0001.

**Figure 5 cells-14-00496-f005:**
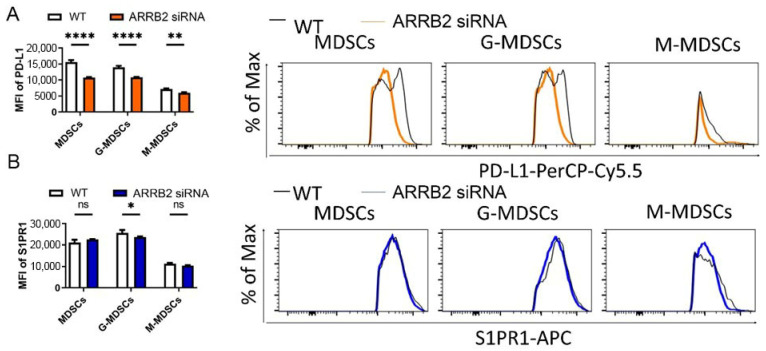
ARRB2 knockdown decreases PD-L1 expression in total MDSCs, G-MDSCs, and M-MDSCs. (**A**) Total MDSCs, G-MDSCs, and M-MDSCs derived from mouse bone marrow cells and subjected to ARRB2 siRNA knockdown exhibited significantly reduced PD-L1 expression, as determined by flow cytometry measuring mean fluorescence intensity (MFI) (**left**), and representative flow cytometry histograms (**right**). (**B**) Among the subsets, only G-MDSCs—but not total MDSCs or M-MDSCs—demonstrated a significant decrease in S1PR1 expression following ARRB2 siRNA knockdown. These findings were assessed by flow cytometry measuring MFI (**left**), with representative histograms shown (**right**). * *p* < 0.05, ** *p* < 0.01, **** *p*< 0.0001.

**Table 1 cells-14-00496-t001:** Clinical characteristics of patients with multiple myeloma.

	High ARRB2 (%) (*n* = 25)	Low ARRB2 (%) (*n* = 22)	*p*-Value
Mean Age	59.8	55.9	0.25
Gender			
Male	11 (44)	11 (50)	0.77
Female	14 (56)	11 (50)	
Race			
Caucasian	15 (60)	14 (63.6)	>0.99
African American	10 (40)	8 (36.4)	
MM Subtype			
IgA-K	4 (16)	3 (13.6)	0.81
IgA-L	3 (12)	5 (22.7)	
IgG-K	11 (44)	9 (40.9)	
IgG-L	2 (8)	4 (18.2)	
Lambda LC	3 (12)	1 (4.5)	
Kappa LC	1 (4)	0	
Unknown/Biclonal	1 (4)	0	
Cytogenetics			
Standard risk	19 (76)	15 (68.2)	0.89
High risk	3 (12)	1 (4.5)	
intermediate	2 (8)	3 (13.6)	
unknown	1 (4)	3 (13.6)	
ISS stage			
1	4 (16)	6 (27.3)	0.78
2	5 (20)	3 (13.6)	
3	8 (32)	7 (31.8)	
Unknown	8 (32)	6 (27.3)	
Response			0.0293
CR	4 (16)	6 (27.3)	
VGPR	6 (24)	4 (18.2)	
PR	4 (16)	8 (36.4)	
SD	0	2 (9.1)	
PD	6 (24)	1 (4.5)	
Unknown	5 (20)	1 (4.5)	
IMiD			
Lenalidomide	19 (76)	21 (95)	0.64
Pomalidomide	2 (8)	6 (27)	
Thalidomide	2 (8)	3 (13)	
Unknown	5 (20)	0	
PI			
Bortezomid	20 (80)	19 (100)	0.72
Carfilzomib	4 (16)	4 (21)	
Ixazomib	1 (4)	0	
Unknown	5 (20)	0	
HSCT			
Autologous	8 (32)	15 (68)	0.042
Tandem-Auto	1 (4)	0 (0)	
None	12 (48)	6 (27)	
Unknown	4 (16)	1 (4)	
Overall response rate	56	81.8	0.0293
Median survival (months)	86.4	107.3	0.0301
Progression-free survival (months)	22	30.13	0.0385

## Data Availability

The data presented in this study are available on request from the corresponding author.

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
