# Peer review of "β-Arrestin 2 as a Prognostic Indicator and Immunomodulatory Factor in Multiple Myeloma"

_cells, 2025, doi:10.3390/cells14070496_

Round 1
Reviewer 1 Report
Comments and Suggestions for Authors
The manuscript by Mathews at al. demonstrates the role of β-arrestin 2 (ARRB2) expression in the pathogenesis and prognosis of multiple myeloma, and show ARRB2 involvement in the PD-1/PD-L1 immune checkpoint pathway regulation.
The authors demonstrate that higher expression of the protein β-arrestin 2 (ARRB2) is associated with poorer clinical outcomes in multiple myeloma patients, and that ARRB2 regulates the expression of the immune checkpoint proteins PD-1 and PD-L1, which may contribute to the pathogenesis of multiple myeloma.
They emphasize also that ARRB2 expression is associated with poor prognosis in multiple myeloma.
The manuscript is well written, well designed, and presents data that are important for understanding the pathobiology of multiple myeloma and response to therapy, as well as immunotherapy and resistance to PD-1/PD-L1 blockade.
Question
Have you checked the serum soluble PD-L1 (sPD-L1) levels?
It has been shown that serum levels of soluble programmed death ligand 1 predict treatment response and progression free survival in multiple myeloma patients.
The English language is clear and easy to understand for the presented data.
Author Response
#Reviewer 1:
The manuscript by Mathews at al. demonstrates the role of β-arrestin 2 (ARRB2) expression in the pathogenesis and prognosis of multiple myeloma, and show ARRB2 involvement in the PD-1/PD-L1 immune checkpoint pathway regulation.
The authors demonstrate that higher expression of the protein β-arrestin 2 (ARRB2) is associated with poorer clinical outcomes in multiple myeloma patients, and that ARRB2 regulates the expression of the immune checkpoint proteins PD-1 and PD-L1, which may contribute to the pathogenesis of multiple myeloma.
They emphasize also that ARRB2 expression is associated with poor prognosis in multiple myeloma.
The manuscript is well written, well designed, and presents data that are important for understanding the pathobiology of multiple myeloma and response to therapy, as well as immunotherapy and resistance to PD-1/PD-L1 blockade.
Question:
Have you checked the serum soluble PD-L1 (sPD-L1) levels?
It has been shown that serum levels of soluble programmed death ligand 1 predict treatment response and progression free survival in multiple myeloma patients.
The English language is clear and easy to understand for the presented data.
Response:
We appreciate the reviewer’s insightful suggestion. Unfortunately, we did not have the opportunity to measure serum soluble PD-L1 (sPD-L1) levels in our study. This was a retrospective study and we did not have biobanked blood samples of these patients. However, we acknowledge the importance of this parameter, as previous studies have demonstrated its potential role in predicting treatment response and progression-free survival in multiple myeloma patients. This would be an interesting aspect to explore in our future research.
Reviewer 2 Report
Comments and Suggestions for Authors
The manuscript entitled: “β-arrestin 2 as a prognostic indicator and immunomodulatory factor in multiple myeloma (ID: cells-3524843)” by Mathews et al. aims to analyze the correlation between ARRB2 expression levels in newly diagnosed MM bone marrow biopsy samples, its impact on outcome and its potential effects in the regulation of the PD-1/PD-L1 checkpoints in murine immune cells.
Albeit the paper is well written and of special interest, comments should be addressed to further improve the manuscript.
Comments:
- Methods section: please provide more insights about the performed “in-silico” analyses in this section.
- Methods section: please highlight more intensively at what timepoint the bone marrow samples of the patients were taken. It would be also of interest if ARRB2 levels were tested during course of the disease.
- Table 1: please provide also for response, IMID, PIs and ASCT p-values where appropriate.
- Discussion section: this section should be more balanced according to strengths and weaknesses of the study. In addition to the authors statement that their results should be validated in a larger cohort, the authors should provide potential recommendation for the clinicians e.g. treatment allocation and which patients should be tested routinely in the future.
Author Response
#Reviewer 2:
The manuscript entitled: “β-arrestin 2 as a prognostic indicator and immunomodulatory factor in multiple myeloma (ID: cells-3524843)” by Mathews et al. aims to analyze the correlation between ARRB2 expression levels in newly diagnosed MM bone marrow biopsy samples, its impact on outcome and its potential effects in the regulation of the PD-1/PD-L1 checkpoints in murine immune cells.
Albeit the paper is well written and of special interest, comments should be addressed to further improve the manuscript.
Comments:
- Methods section: please provide more insights about the performed “in-silico” analyses in this section.
Response:
Thanks for your comments. We added a new section for this datasets analysis in the Methods (please see line 158 to 163 in the revised manuscript). Briefly, we downloaded the dataset and analyzed using the http://www.genomicscape.com/.
- Methods section: please highlight more intensively at what timepoint the bone marrow samples of the patients were taken. It would be also of interest if ARRB2 levels were tested during course of the disease.
Response:
We have clearly indicated in the revised manuscript that all bone marrow samples were collected from newly diagnosed multiple myeloma patients at the time of initial diagnosis. We acknowledge the importance of assessing ARRB2 levels at different time points and in relation to disease resistance, which would be valuable for future studies (please see line 458 to 460 in the revised manuscript).
- Table 1: please provide also for response, IMID, PIs and ASCT p-values where appropriate.
Response:
We have added the p-values as requested. Please refer to the updated version of the manuscript for the revised Table 1.
- Discussion section: this section should be more balanced according to strengths and weaknesses of the study. In addition to the authors statement that their results should be validated in a larger cohort, the authors should provide potential recommendation for the clinicians e.g. treatment allocation and which patients should be tested routinely in the future.
Response:
This is an excellent suggestion. We have included a statement in our Discussion section that “our findings have important clinical relevance: measurement of ARRB2 in bone marrow biopsy samples could help risk stratification of NDMM patients and for patients with high ARRB2 expression, more intensive chemotherapy will be needed”. (Please see line 424 to 427 in the revised manuscript)
Reviewer 3 Report
Comments and Suggestions for Authors
The authors report that higher levels of ARRB2, a member of the arrestin family, is associated with inferior PFS and OS and that its expression is associated with PD1 expression.
My notes:
-- I believe that the problem with PD1/PD-L1 blockade and IMiDs was that the combination was too toxic, which should be mentioned in the text.
--Were the 47 patients studied consecutive? Or were there some excluded based on certain criteria?
--Please check with a statistician that p values were appropriately used in the "patient characteristics" part of Table 1.
--In the Figure 1B legend, I would note what defined high and low ARRB2 expression.
--Since PD-L1/PD1 blockade is not typically used in multiple myeloma, why do you think it is that ARRB2 levels are associated with differences in PFS and OS?
--In Figure 3A, the difference between WT and KO is only present at Day 3; levels also drop in WT by day 7. Why do you think that is?
Author Response
#Reviewer 3:
The authors report that higher levels of ARRB2, a member of the arrestin family, is associated with inferior PFS and OS and that its expression is associated with PD1 expression.
My notes:
-- I believe that the problem with PD1/PD-L1 blockade and IMiDs was that the combination was too toxic, which should be mentioned in the text.
Respone:
We agreed that the combination of PD-1/PD-L1 blockade and IMiDs was too toxic. We have clarified this in our revised manuscript (Lines 111-112 of the revised manuscript), stating that " the combination was felt to be overly toxic with a high burden of significant adverse events” Additionally, we have cited references 38 and 39 to support this statement.
--Were the 47 patients studied consecutive? Or were there some excluded based on certain criteria?
Respone:
The patients were not enrolled consecutively; inclusion depended on the availability of biopsy samples at our institution. I have clarified this in our revised manuscript (Line 144-145).
--Please check with a statistician that p values were appropriately used in the "patient characteristics" part of Table 1.
Respone:
We have added the p-values as requested. Please refer to the updated version of the manuscript for the revised Table 1.
--In the Figure 1B legend, I would note what defined high and low ARRB2 expression.
Respone:
Yes. I have defined high and low ARRB2 expressioin in Fig 1B legend (please see line 301-302 in the revised manuscript)
--Since PD-L1/PD1 blockade is not typically used in multiple myeloma, why do you think it is that ARRB2 levels are associated with differences in PFS and OS?
Respone:
Excellent question. PD-L1/PD1 blockade is not typically used in multiple myeloma as the use of PD-L1/PD1 blockade in myeloma was associated with increased risks of neutropenic sepsis, myocarditis and Stevens-Johnson syndrome. Nevertheless, PD-1/PD-L1 signaling remains an important pathway in regulating tumor immunity and it is important to show that ARRB2 plays an important role in PD-1/PD-L1 regulation. However, there are likely multiple other mechanisms contributing to the association between ARRB2 levels and differences in PFS and OS beyond PD-1/PD-L1 alone. We have added a statement that in addition to PD-1 and PD-L1, ARRB2 plays important roles in the regulation of many other molecules/receptors that are critical to MM pathogenesis (Line 494-496)
--In Figure 3A, the difference between WT and KO is only present at Day 3; levels also drop in WT by day 7. Why do you think that is?
Respone:
We think this is likely that at Day 7 of continuous in vitro stimulation and activation the majority of T cells have undergone cell death, which may explain the difference in PD-1 expression between WT and KO T cells was apparent at Day 3 but not at day 7. Further studies are needed to better understand the dynamics of cell viability and the mechanisms underlying this time-dependent effect.
Reviewer 4 Report
Comments and Suggestions for Authors
In this manuscript, the authors investigated the association between ARRB2 expression and clinical outcomes in patients with multiple myeloma (MM) and examined the role of ARRB2 in the regulation of PD-1/PD-L1. This study demonstrated that higher ARRB2 expression in the bone marrow of newly diagnosed MM patients was associated with inferior progression-free survival and overall survival. Additionally, ARRB2 knockdown led to the downregulation of PD-1 expression in murine CD3 T cells and PD-L1 expression in murine myeloid-derived suppressor cells. The study presents novel findings, is methodologically sound, and is well-supported by experimental results. I recommend acceptance after addressing the following minor revisions:
- In Figure 1B, please indicate which curve represents high ARRB2 expression levels and which represents low ARRB2 expression levels.
- In the section “Higher ARRB2 expression correlates with disease progression and poor survival in newly diagnosed myeloma (NDMM) patients”, please include a description of how high and low ARRB2 expression levels were defined.
- In Figure 3A, why does PD-1 expression in WT splenic T cells decrease along with days after activation?
- In Figure 4A, why does ARRB2 appear as multiple bands? Please quantify the bands and normalize them.
- In Figure 4C, the flow cytometry histograms for CD4+ and CD8+ T cells exhibit multiple peaks or irregular patterns. Since CD4+ and CD8+ T cells are homogeneous cell populations, their distributions should be more uniform, like normal distribution. Please clarify this observation.
- Please include the gating strategy for MDSCs using CD11b, Ly6C, and Ly6G antibodies in the Methods section.
- Please define G-MDSCs and M-MDSCs in the manuscript.
- In Figures 4B and 5B, why is the decrease in S1PR1 expression in MDSCs less pronounced than in T cells?
Author Response
# Reviewer 4:
In this manuscript, the authors investigated the association between ARRB2 expression and clinical outcomes in patients with multiple myeloma (MM) and examined the role of ARRB2 in the regulation of PD-1/PD-L1. This study demonstrated that higher ARRB2 expression in the bone marrow of newly diagnosed MM patients was associated with inferior progression-free survival and overall survival. Additionally, ARRB2 knockdown led to the downregulation of PD-1 expression in murine CD3 T cells and PD-L1 expression in murine myeloid-derived suppressor cells. The study presents novel findings, is methodologically sound, and is well-supported by experimental results. I recommend acceptance after addressing the following minor revisions:
- In Figure 1B, please indicate which curve represents high ARRB2 expression levels and which represents low ARRB2 expression levels.
Response:
This has been addressed. In the revised figure, we have clearly indicated which curve represents high ARRB2 expression levels and which represents low ARRB2 expression levels.
- In the section “Higher ARRB2 expression correlates with disease progression and poor survival in newly diagnosed myeloma (NDMM) patients”, please include a description of how high and low ARRB2 expression levels were defined.
Response:
We have defined high and low ARRB2 expression in the Methods section under ‘Immunofluorescent staining of bone marrow ARRB2 and quantification,’ stating: “ARRB2 expression is categorized as high if the index number is ≥1 and low if the index number is <1.” (Line 215-216 in the revised manuscript). The definition of high and low ARRB2 expression levels were emphasized in Results section under ‘Higher ARRB2 expression correlates with disease progression and poor survival in newly diagnosed myeloma (NDMM) patients,’ stating: ‘The quantitative level of ARRB2 in each bone marrow biopsy sample was assessed using ARRB2 expression index number: (A2CT signal from sample – A2CT signal from IgG control)/DAPI. ARRB2 expression is categorized as high if index number is ³ 1 or low if index number is <1 (Line 309-312 in the revised manuscript).
- In Figure 3A, why does PD-1 expression in WT splenic T cells decrease along with days after activation?
Response:
We think the decrease in PD-1 expression along with days after activation is likely due to cell death. This is likely that at Day 7 of continuous in vitro stimulation and activation the majority of T cells have undergone cell death.
- In Figure 4A, why does ARRB2 appear as multiple bands? Please quantify the bands and normalize them.
Response:
The antibody we used for the measurement of ARRB2 was generated in house at Dr. Robert Lefkowitz lab. The multiple bands observed for ARRB2 were likely due to backgound staining. The ARRB2 band has now been quantified and normalized in the revised manuscript (See the right panel in the revised Figure 4A.
- In Figure 4C, the flow cytometry histograms for CD4+ and CD8+ T cells exhibit multiple peaks or irregular patterns. Since CD4+ and CD8+ T cells are homogeneous cell populations, their distributions should be more uniform, like normal distribution. Please clarify this observation.
Response:
The flow cytometry histograms shown in Figure 4C represent PD-1 and S1PR1 expression levels, gated on the CD4+ and CD8+ T cell populations, respectively, and not the CD4/CD8 cell number or CD4/CD8 expression level. The observed variability and multiple peaks in the histograms reflect the heterogeneous expression of PD-1 and S1PR1 across individual cells. PD-1 and S1PR1 expression varies depending on the cellular status. This explains the irregular patterns seen, as the expression of these markers is not uniform within the CD4/CD8 cell populations.
- Please include the gating strategy for MDSCs using CD11b, Ly6C, and Ly6G antibodies in the Methods section.
Response:
The gating strategy for MDSCs has been added to the Methods section (please see line 244 to 247 in the revised manuscript).
- Please define G-MDSCs and M-MDSCs in the manuscript.
Response:
We have defined G-MDSCs (granulocytic MDSCs) and M-MDSCs (monocytic MDSCs) in the manuscript (line 246-247 in the revised manuscript).
- In Figures 4B and 5B, why is the decrease in S1PR1 expression in MDSCs less pronounced than in T cells?
Response:
S1PR1 expression is higher in T cells compared to MDSCs at baseline. This disparity in initial expression levels likely accounts for the difference in the observed differences of S1PR1 between T cells and MDSCs in ARRB2 depleted cells.